# Analysis of Entropy Generation for Mass and Thermal Mixing Behaviors in Non-Newtonian Nano-Fluids of a Crossing Micromixer

**DOI:** 10.3390/mi15111392

**Published:** 2024-11-17

**Authors:** Ayache Lakhdar, Jribi Skander, Naas Toufik Tayeb, Telha Mostefa, Shakhawat Hossain, Sun Min Kim

**Affiliations:** 1Laboratory of Electro-Mechanical Systems, The Engineers National School of SFAX, University of SFAX, Sfax 3038, Tunisia; lakhdar.ayache@enis.tn (A.L.);; 2Department of Mechanical Engineering, College of Engineering, King Faisal University, Al-Ahsa 36362, Saudi Arabia; 3Laboratory of Renewable Energy Systems Applications, Gas Turbine Joint Research Team, Ziane Achour University, Djelfa 17000, Algeria; t.naas@univ-djelfa.dz; 4Department of Mechanical Engineering, Ziane Achour University, Djelfa 17000, Algeria; m.telha@univ-djelfa.dz; 5Department of Industrial and Production Engineering, Jashore University of Science and Technology, Jessore 7408, Bangladesh; 6Department of Mechanical Engineering, Inha University, 100, Inha-ro, Michuhol-gu, Incheon 22212, Republic of Korea; 7Department of Biological Sciences and Bioengineering, Inha University, 100, Inha-ro, Michuhol-gu, Incheon 22212, Republic of Korea; 8Biohybrid Systems Research Center, Inha University, 100, Inha-ro, Michuhol-gu, Incheon 22212, Republic of Korea

**Keywords:** TLCC micromixer, chaotic advection, mixing energy cost, entropy generation and irreversibilities

## Abstract

This work’s objective is to investigate the laminar steady flow characteristics of non-Newtonian nano-fluids in a developed chaotic microdevice known as a two-layer crossing channels micromixer (TLCCM). The continuity equation, the 3D momentum equations, and the species transport equations have been solved numerically at low Reynolds numbers with the commercial CFD software Fluent. A procedure has been verified for non-Newtonian flow in studied geometry that is continuously heated. Secondary flows and thermal mixing performance with two distinct intake temperatures of nano-shear thinning fluids is involved. For an extensive range of Reynolds numbers (0.1 to 25), the impact of fluid characteristics and various concentrations of Al_2_O_3_ nanoparticles on thermal mixing capabilities and pressure drop were investigated. The simulation for performance enhancement was run using a power-law index (*n*) at intervals of different nanoparticle concentrations (0.5 to 5%). At high nano-fluid concentrations, our research findings indicate that hydrodynamic and thermal performances are considerably improved for all Reynolds numbers because of the strong chaotic flow. The mass fraction visualization shows that the suggested design has a fast thermal mixing rate that approaches 0.99%. As a consequence of the thermal and hydrodynamic processes, under the effect of chaotic advection, the creation of entropy governs the second law of thermodynamics. Thus, with the least amount of friction and thermal irreversibilities compared to other studied geometries, the TLCCM arrangement confirmed a significant enhancement in the mixing performance.

## 1. Introduction

Among the best working passive mixing strategies for enhancing flow mixing is the chaotic advection strategy. The two-layer micromixer is a potentially chaotic geometry that could provide a viable approach to enhance fluid kinematics and hydrodynamic performance in a 3D laminar steady flow. Many designs for applying chaotic advection to passively increase fluid mixing have been provided empirically and numerically by some researchers who have employed such geometry in their experiments [1,2,3,4,5]. Their results show that when the Reynolds number rises, the micromixer’s mixing rate increases.

Various micromixers characteristics find extensive use in various industrial contexts [6,7,8]. In certain processes, mixing is crucial in the laminar regime at a low Reynolds number. As a well-proven technique of enhancing mixing efficiency involves the utilization of chaotic advection [9], the secondary flows that arise from it are quite strong and influence homogenization at the microscopic level [10,11,12].

Growing interest in the topic of micromixing has been observed in recent years, which involves the efficient blending of fluids on a small scale. This emerging field is driven by the need for the accurate control and optimization of chemical reactions, as well as the desire to reduce resource consumption and waste production [13]. This has led to the improvement of innovative micromixers that aim to overcome the challenges posed by laminar flow in microchannels [14,15].

Limited progress has been made in improving hydrodynamic and heat flux despite insufficient flow mixing. Several research teams have used various advanced geometries to increase the mixing hydrodynamic and heat transfer rate [16,17,18,19,20,21,22].

Furthermore, the use of nano-fluids has been explored as a means to improve the thermal mixing efficiency of micromixers studied by Huminicet al. [23]. The implementation of nano-fluids in micromixers offers an alternative to conventional thermal systems and has the potential to significantly enhance their mixing efficiency. The literature review suggested that the use of nano-fluids in micromixers could be a great substitution for traditional thermal systems and an interesting topic for further research.

The incorporation of non-Newtonian nano-fluids, such as single-walled carbon nanotubes–engine oil (SWCNT-EO) or molybdenum disulfide–polyethylene glycol Casson nano-fluid, in micromixers represents a cutting-edge development in enhancing thermal efficiency and minimizing pressure drops [23]. Understanding the behavior and characteristics of non-Newtonian nano-fluids in micromixers is crucial for the engineering design and optimization of different processes [24,25]. Hossainet al. [26,27] numerically illustrated the mixing of two different fluids, water and dyewater, within an OH-shaped microstructure using CDF software of version 2016^y^. Then, utilizing CFD code, they proposed a novel complicated shape known as the SAR micromixer. Therefore, non-Newtonian nano-fluids exhibit unique rheological properties due to the presence of nanoparticles, which can significantly affect their flow pattern efficiency and mixing performance.

Researchers have experimentally improved different applied thermal engineering systems [28,29] and numerically [30,31,32,33,34,35,36,37] to increase heat transfer efficiency. They have employed efficient nanoparticles such CuO [35,36,37], H_2_O/SWCNT [35], and Al_2_O_3_ [30,31]. In comparison to an identical flow with 0% nano-concentration, the authors’ experimental results demonstrate that the thermal conductivities of the nano-fluids are much stronger. Xuan and Li [38] conducted an experimental investigation on the thermal performance of nano-fluid whirling under wall heat transfer inside tubes. At a Cu nanoparticle concentration of 2.0%, the heat transfer coefficient to pure water increased by over 39%. Esmaeilnejad et al. [39] studied the laminar convection flow system due to nano-shear within rectangular microchannels. According to their obtained results, the pressure drop increases by about 50.7% and the thermal coefficient decreases by approximately 27.2% at a particle concentration of 4% and a Peclet number of 700. Karvelas et al. [40] conducted research on the ability of heated water to mix when subjected to an electromagnetic field. Pouya [41] looked at the mixing quality and heat transfer enhancement of a hybrid nano-fluid numerically for various mixers. They discovered that the mixing rate grew with time for high Reynolds numbers and constant frequencies. Increasing the heat transfer and lowering the pressure drop, regarding entropy formation, numerous research teams employed numerically complex geometries [42,43,44,45,46]. The entropy generation rate calculates the amount of energy lost in a system as a result of irreversibility. The amount of entropy generated can be significantly influenced by a channel’s geometry. Moreover, the study of entropy generation in micromixers is crucial for understanding the efficiency of mixing processes at the microscale. Entropy generation is a significant factor in assessing the overall performance of micromixers, as it reflects the irreversibility of mixing processes and the associated energy losses [47], especially when dealing with non-Newtonian fluids. By investigating the entropy generation in micromixers for non-Newtonian fluids, researchers can develop strategies to enhance mixing efficiency and reduce energy consumption, contributing to the advancement of micro-fluidic technology [48].

Furthermore, various studies [49,50,51] have highlighted the significant impact of nanoparticle concentration on entropy production in non-Newtonian fluid flows, indicating an optimal volume fraction for minimizing irreversibility. By integrating these findings into the evaluation of non-Newtonian nano-fluids in micromixers, a comprehensive understanding of the interplay between fluid properties, heat transfer efficiency, and entropy generation can be achieved.

A review provides insights into the fundamental mixing mechanisms of non-Newtonian fluids in microscale channels, focusing on nano-enhanced fluids [52].

Various studies explain how different passive micromixer designs influence mixing efficiency when using non-Newtonian nano-fluids, providing comparative data on mixing performance across designs [53,54,55,56]. Researchers have explored mixing characteristics in biomedical applications, focusing on non-Newtonian nano-fluids and addressing the interaction between fluid properties and micromixer design.

In this research, we aim to examine the formation of entropy and the efficiency of thermal mixing for a non-Newtonian nano-fluid within a novel micromixer referenced in [25]. In recent years, micromixing has attracted considerable interest due to the need for the accurate control and optimization of chemical reactions, along with efforts to decrease resource usage and waste generation. Furthermore, the incorporation of nano-fluids is viewed as a promising strategy to improve thermal mixing efficiency and reduce entropy production in micromixers, providing a viable alternative to conventional thermal systems. Through this research, we aim to contribute to the understanding of nano-fluid-based micromixers and assess their potential as an alternative to traditional thermal systems, thus paving the way for further exploration in this domain. The suggested micromixer’s chaotic flow generation and thermal mixing performances were examined using a range of nanoparticle concentrations and fluid behavior index values. The evaluation of the mixing energy cost and fluid index homogenization will be conducted to obtain significant energy efficiency and minimize the entropy generation. This research aims to enhance the understanding of micromixers that utilize nano-fluids and evaluate their viability as alternatives to conventional thermal systems, thereby encouraging further investigation in this area. We analyzed the chaotic flow generation and thermal mixing performance of the proposed micromixer by varying nanoparticle concentrations and fluid behavior index values. Additionally, we will assess the mixing energy costs and the homogenization of the fluid index to achieve improved energy efficiency and reduced entropy generation.

## 2. Materials and Methods

### 2.1. Problem Synopsis and Micromixer Design

In this study, a novel micromixer, a modified two-layer crossing geometry (TLCM), was proposed. It was initially applied by Naas et al. [16] to accomplish higher mixing performance for power-law non-Newtonian fluids under the influence of nano-fluid concentrations [57]. Figure 1 illustrates the TLCM geometry.

Two twisted channels make up the micromixer; a periodic chamber is made in the arrangement of the lower and upper channels. The following mixing units are found in multiple grooves that have been recreated. Table 1 displays the specific dimensions; d represents the groove diameter, I is the distance among the inlets, D is the chamber diameter, d_hyd_ is the hydraulic diameter, and L^*^ the geometry’s length. Table 1 and Table 2 show the non-Newtonian nano-fluid that was suggested.

Different flow temperatures were proposed in the inlet sections with a constant velocity. In addition to the no-slip borders, the outsides are considered adiabatic. The outflow segment is where the pressure outlet condition is awarded. The formulae that follow represent the governing equations [25,26] and were numerically solved by a CFD program:
(1)$$\mathrm{div}\vec{V}=0$$
where $\vec{V}$ represents the velocity vector.
(2)$$\overset{⃑}{V}.\overset{̿}{\nabla }\overset{⃑}{V}=-\frac{1}{\rho_{nf}}\vec{\nabla }P+div\tau$$
where τ (Pa) and *P* represent the shear stress and pressure, correspondingly.
(3)$$\rho_{nf}c_{nf}\overset{⃑}{V}.\vec{\nabla }T=\lambda_{nf}\Delta T$$
where $\lambda_{nf}$, *T* and $\rho_{nf}$ are the conductivity, temperature, and density of the working nano-fluid studied in this literature, respectively. A basic power-law equation can be used to describe the constitutive relationship between the shear rate *γ* (s^−1^) and shear stress *τ* (Pa), as follows:(4)$$\tau =m{\dot{\gamma }}^{n}$$
where *n* and *m* represent the fluid behavior index and fluid consistency index.

The viscosity equation can be mathematically articulated by the following:(5)$$\mu_{nf}=k{\dot{\gamma }}^{n-1}$$

The following boundary conditions are suggested:A consistent velocity profile applied to the inlets’ flow.At the inlets, the maximum and minimum temperatures are set at 300 and 330, respectively.The solid walls have non-slip properties.The output section flow takes the pressure outlet condition into account.

### 2.2. Properties of Mass Transfer in Chaotic Flows

For power-law nano-fluids, the Reynolds number (*Re*) was defined as follows by Metzner and Delplace [58,59] and used by Tayeb et al. [25]:(6)$$Re=\frac{\rho_{nf}u^{2-n}d_{hyd}^{n}}{\left[8^{n-1}{\left(b^{*}+\frac{a*}{n}\right)}^{n}m\right]}$$

The constant geometric parameters, *a** and *b**, have values of 0.6771 and 0.2121, respectively. The mixing index is developed as follows in order to compare the micromixers’performances [25,59]:(7)$$MI=1-\frac{\sigma }{\sigma_{0}}$$

The standard deviation of the mass fraction is denoted as follows:(8)$$\sigma^{2}=\frac{1}{N}\sum_{i=1}^{N}{\left(C_{i}-\bar{C}\right)}^{2}$$
where *N* is the total number of sampling points in the transversal segment, the mass fraction at the inspection point *i* is denoted by $C_{i}$, the ideal mixing mass fraction is denoted by $\bar{C}$, and the inlet section $\sigma_{0}$ represents the standard deviation (SD). The maximum SD for the data range is found using the formula below:(9)$$\sigma_{0}^{2}=\bar{C}\left(1-\bar{C}\right)$$

The evolutions of the mean vorticity rate ($\Omega_{mean})$ in the micromixer as function of the Reynolds number ranging from 0.5 to 25. This parameter is defined by the following equations [18]:(10)$$\Omega =\frac{1}{2}{\left[{\left(\frac{\partial w}{\partial y}-\frac{\partial v}{\partial z}\right)}^{2}+{\left(\frac{\partial u}{\partial z}-\frac{\partial w}{\partial x}\right)}^{2}+{\left(\frac{\partial v}{\partial x}-\frac{\partial u}{\partial y}\right)}^{2}\right]}^{\frac{1}{2}}$$
(11)$$\Omega_{mean}=\frac{1}{℧}\int \Omega d℧$$
where ℧ represents the total volume of the fluid in the channel.

Better flow rates and energy consumption are correlated with a greater mass mixing index. The estimation process needs to strike a compromise between the cost of mixing and its input power efficiency (the energy required to move the fluid down the channel) [59]:(12)$$MMEC=\frac{\Delta P\times Q}{MI}$$

### 2.3. Properties of Heat Transfer in Chaotic Flows

With varying inlet temperatures, the coefficient of heat transfer, *h*, is expected to be as follows:(13)$$h=\frac{q^{\prime \prime }}{\left(T_{b}-T_{W}\right)}$$
where the heat flux of the wall is expressed as *q*″ (w/m^2^), the average wall temperature is defined as *T_w_* (*k*), and the mean bulk temperature is called *T_b_*(*k*). Thermal mixing is measured by the thermal mixing index (TMI), which is calculated as follows for hot and cold fluids.
(14)$$TMI=1-\frac{\sqrt{\frac{1}{n}\sum_{i=1}^{n}\;{\left(T_{i}-\bar{T}\right)}^{2}}}{\sigma_{0}}$$
where ($\bar{T}$) is the average temperature at the chosen plane and on node I, and the average temperature is denoted by *T_i_*. Increasing flow rates and energy consumption are correlated with best mixing indices. They are defined as follows [25] and must be evaluated to strike a balance between the cost of thermal mixing and its value efficiency regarding input power (the inertia required to push the fluid):(15)$$TMEC=\frac{\Delta P\times Q}{TMI}$$

Using the flow field’s temperatures and velocity distribution, one may ascertain the irreversibility of the local entropy generation due to heat transfer ($s_{T}^{\prime \prime \prime }$) and the irreversibility of fluid friction ($s_{P}^{\prime \prime \prime }$) in three dimensions of flow [48,60,61]:(16)$$s_{T}^{\prime \prime \prime }=\frac{\lambda }{T^{2}}\left[{\left(\frac{\partial T}{\partial x}\right)}^{2}+{\left(\frac{\partial T}{\partial y}\right)}^{2}+{\left(\frac{\partial T}{\partial z}\right)}^{2}\right]$$
(17)$$s_{P}^{\prime \prime \prime }=\frac{\mu }{T}\left[2\left({\left(\frac{\partial u}{\partial x}\right)}^{2}+{\left(\frac{\partial v}{\partial y}\right)}^{2}+{\left(\frac{\partial w}{\partial z}\right)}^{2}\right)+{\left(\frac{\partial u}{\partial y}+\frac{\partial v}{\partial x}\right)}^{2}+{\left(\frac{\partial u}{\partial z}+\frac{\partial w}{\partial x}\right)}^{2}+{\left(\frac{\partial v}{\partial z}+\frac{\partial w}{\partial y}\right)}^{2}\right]$$

It is possible to calculate the total generation of the entropy within the fluid flow, which aims to provide a brief explanation of fluid homogenization, in the following way:(18)$$s_{gen}^{\prime \prime \prime }=s_{T}^{\prime \prime \prime }+s_{P}^{\prime \prime \prime }$$

The ratio of thermal to total losses is computed using the Bejan number [48,62]:(19)$$Be=\frac{S_{T}^{\prime \prime \prime }}{S_{gen}^{\prime \prime \prime }}$$

### 2.4. Numerical Approach, Testing of Mesh Sensitivity, and Validations

With (FVM) finite volumes method-based ANSYS Fluent 16© CFD software [63], all of the governing equations in this work were resolved in a laminar flow regime. For the coupling of pressure and velocity, the SIMPLEC scheme was chosen. The mass and momentum equations were found using a second-order upwind technique. It was confirmed and simulated that the calculations would converge at 10^−7^ root mean square (RMS) residual values. As working fluids, non-Newtonian power-law fluids have been used for a range of Al_2_O_3_ nanoparticle concentrations.

By altering the total number of cells, a quantitative grid test was conducted to assess the sensitivity of the numerical outcomes. Four mesh grids, ranging from 100,000 to 800,000 nodes, were examined using unstructured mesh with homogeneous tetrahedral cells of the mesh sizes 0.001 to 0.00011 mm; see Figure 2 and Figure 3.

The mixing index, which measures the efficiency of the mixing, was evaluated with mass transfer distribution at the different outlet sections with an increasing number of mesh cells, as shown in Figure 4. Also, Table 3 presents the pressure drop and standard deviation of fluids to understand the mesh independent test. According to the mesh sensitivity results, a grid of 600,000 cells corresponding to a mesh size of 11 μm was chosen.

To confirm the accuracy of the numerical code, numerical simulations were confirmed using a chaotic micromixer with obstacles [5] at a fixed Reynolds number, as shown in Table 4. The difference in inaccuracy between Chia et al.’s results and our simulations was found to be significantly less than 1%.

Additionally, a quantitative numerical validation was conducted using the data from Li et al. [64], and the findings show the heat transfer rate for non-Newtonian instances as a function of different Reynolds numbers. Table 5 displays the results of an acceptable comparison that showed good agreements among the outcomes.

## 3. Results and Discussion

This study delves deeply into the kinematic and thermal characteristics of the mixing of non-Newtonian nano-fluids in a novel micromixer. Fluid mixing processes, mass transfer, and the second law of thermodynamics are examined for a wide variety of low Reynolds numbers, from 0.1 to 40.

### 3.1. Mass Transfer and Fluid Mixing Processing

Figure 5 displays the flow characteristics of the mass fraction among nano-fluids. The fluid pattern is treated to various scenarios of fluid concentrations in order to comprehend the evolution of visual blending in the new setup. We observe that, for all nano-fluid cases, the fluid mixing increases with the *Re*. The new micromixers perform quickly in terms of mass transfer as a result.

For nano-fluids, kinematic behavior plays a significant role in improving homogeneity. The micromixer has a low-pressure drop close to the outlet portion, as shown in Figure 6, but it also features a single powerful vortex zone within each corner that increases the mixing rate. Furthermore, the configuration’s structure and curvature allow us to observe that the flow is more chaotic and dynamic, allowing for transfer effectiveness and guaranteeing a superior homogenization quality.

In addition, it is noteworthy that the path-line within the chosen new micromixer generates a strong secondary flow and a reversed flow pattern, improving not only the mass transfer efficiency but also assuring high-quality homogenization.

The fluids’ mass fraction distributions at different concentrations of nano-fluid (ϕ = 0.5 to 5%) are shown in Figure 7. *Re* = 0.5 to 25 represents these distributions at the micromixer’s exit. It is observed that the micromixer’s crossing nodes encourage the growth in stretching and compression within the cross sectional area, which, starting in the fourth phase of the micromixer, leads to more uniform mixing.

The mixing index generally rises for all Reynolds numbers in Figure 8, suggesting that the two fluids become more homogenized and perform significantly better at mixing than the shear thinning fluid with φ = 1%. The diffusion regime at very low Reynolds number leads to much higher mixing index values for the micromixer.

Because of the extremely chaotic advection impact, the mixing of micromixers with a high fluid concentration climbs to 93% for *Re* between 25. Since the nanoparticles alone are able to homogenize the flow, the Reynolds number is very low (*Re* < 5), and the flow behaviors are ineffective in enhancing mixing (molecular diffusion predominates).

As the Reynolds number increases, the homogenization is more effective and the mixing intensity rapidly develops. It should be noted that at high Reynolds numbers, the nanoparticles work better and the mixing intensity builds up faster, allowing the concentration to rise to the most selective mixing state. Furthermore, compared to the situation of n = 0.88, it is seen that the suggested micromixer exhibits a 2.22% boost in mixing intensity when the fluid behavior index drops to 0.46.

Table 6 illustrates the impact of Reynolds numbers on the vortex intensity of fluid in five different nano-fluid cases. As the Reynolds number increases, for all micromixers, it becomes clear that the flow strength also increases, resulting in higher kinetic energy and strong chaotic advection. Consequently, both vorticity and secondary flow develop rapidly with rising Reynolds numbers. For a specific Reynolds number, the dynamic flow is similarly vigorous as φ  decreases, contributing to the chaotic agitation within the present micromixer.

An analysis of the cost of combining energy MEC with several nanoparticle concentration scenarios for varying Reynolds numbers is presented in Figure 9. The projected cost of mixing energy is expressed regarding input power (mW). Because the flow velocity directly affects the rise in the flow rate, it is seen that the mixing energy cost increases for all scenarios as the Reynolds number increases. This is because the flow velocity affects the thermal and hydrodynamic conditions. This is a result of the pressure drop changing at a higher order of magnitude than the mixing index changing as the Reynolds number increases. At low Reynolds numbers, such as *Re* ≤ 10, the mixing energy cost is typically smaller.

However, the mixing energy cost is highly dependent on the power-law index, and its values decrease as the index or fluid concentration increases. This is related to the rheological fluid behaviors, as an increase in the power-law index causes the apparent viscosity to decrease, which in turn allows for easier fluid agitation. As a result, the mixing energy cost drops, and the mixing index rises significantly.

Table 7 provides a comparison of the mass mixing energy cost (MMEC) for the proposed micromixer alongside several recent micromixers across a range of Reynolds numbers.

### 3.2. Heat Transfer and Thermal Mixing Mechanism

For a range of nano-fluid concentrations (ϕ = 0.5 to 5%), thermal mixing fluids within cold and hot non-Newtonian nano-fluids are calculated. Referring to Figure 10, this is accomplished by injecting the heated fluid at 330 K into one inlet and the cold fluid at 300 K into the other.

This section looks into the current micromixer’s mining energy cost and thermal mixing capabilities. Its results are contrasted with those of more powerful micromixers. A top view of the temperature contours for three different nano-fluid concentration scenarios (ϕ = 0, 2.5, and 5%) for a Reg ranging from 0.1 to 25 is presented in Figure 10. When the Reg is more significant, the homogenization quality of the thermal mixing is stronger for a specific value of the fluid behavior index or nano-fluid concentration.

As the number increases, the movement’s dynamic is greatly enhanced, the fluid nanoparticles’ kinematics vary significantly, and the mixing intensity is raised.

The contours of heat mixing for varying fluid concentrations and *Re* are displayed in Figure 11 to explain how heat transport is affected by secondary flux produced by eddies. The figures show that homogeneity plays a significant role in the flow’s eddy zones. We discovered from the results that the mixer is used to assess the thermal performance in every situation. The fluids are thoroughly mixed and tend to homogenize as they move through the geometry in the diffusion regime with *Re* = 0.5 because of the chaotic mechanism of the fluid flow. It is evident that when φ = 5% there is a greater impact on the mixing process and a higher level of homogeneity.

As shown in Figure 12, the thermal mixing index (TMI) is a measurement made in the output fluid flow for different *Re* with the influence of nanoparticle concentrations. It can be defined as the temperature variance divided by the mean temperature. The thermal mixing capacities are enhanced in all *Re* situations; the maximum TMI is achieved at high nano-fluid concentrations, the and TMI approaches 100% (complete mixing).

The ratio of the pressure drop to the mixing efficiency can be used to compute the mixing cost. Not only higher flow rates but also higher energy consumption are generally correlated with higher mixing indices, which increases the mixing cost needed to describe the mixing performance. The energy cost of mixing generally rises with the Reynolds number. In addition, the size and form of the mixing vessel can have an impact on the energy cost of mixing (see Figure 13).

### 3.3. Thermodynamics and Entropy Generation Processing

In this section, we describe the creation of entropy resulting from heat transfer and fluid friction, considering the influence of Reynolds numbers and varying nano-fluid concentrations.

In chemical reactions, minimizing entropy can help in achieving more ordered states, which often corresponds to higher yields of desired products. By optimizing conditions (such as temperature and pressure) to favor lower entropy states, processes can become more efficient. In addition, entropy minimization can be applied in the design of drug delivery systems where the goal is to achieve targeted and controlled release. By creating more ordered structures (like nanoparticles in this work), the stability and efficacy of drug delivery can be enhanced, reducing wastage and improving therapeutic outcomes.

Because laminar flow has a relatively low micromixer geometry, the formation of local frictional entropy is dependent upon it. More entropy is generated as a result of the inertial force becoming more pronounced as the Reynolds values rise. Moreover, as demonstrated in Figure 14, the entropy produced by fluid friction is marginally less in φ = 5% than in other scenarios because of a greater associated pressure decrease. The effects of different amounts of nano-fluid are negligible, though.

A comparison of the entropy generation resulting from heat transfer with varying values of φ and *Re* is shown in Figure 15. Regarding the micromixer, it is evident that in all non-Newtonian fluid instances, the irreversibility of the thermal entropy generation is smaller and its size diminishes with an increase in flow behavior (φ). Heat transfer entropy formation rises with an increasing *Re*. These findings verify that by lowering *Re* and raising n, the chaotic flow can successfully improve thermal performance. Low concentrations of nano-fluid lead to the greatest development of entropy. These findings reaffirm that the suggested chaotic micromixer can significantly improve heat mixing efficiency in terms of irreversible heat flow.

Figure 16 displays the changes in the global entropy generation with φ for various values of *Re*. As previously stated, in every scenario, the formation of total entropy increases when *Re* rises with an increase in φ. The reason for this evolution is that the temperature gradients in the flow increase.

The impact of *Re* on the mean Bejan number for each suggested nano-fluid concentration is shown in Figure 17. It is proven that the mean Bejan number values are greater than 0.8. Therefore, in all non-Newtonian cases, heat transfer irreversibility dominates the generation of entropy. The reason for this is that the gradients of temperature are greater than those of velocity. In general, the fluid friction entropy generation is less significant, which causes the suggested micromixer to have a larger mean Bejan number. Therefore, increased process efficiency would come from the flow with high φ and *Re*.

## 4. Conclusions

In conclusion, the use of non-Newtonian nano-fluids in a novel micromixer has shown promising results in enhancing entropy generation and improving thermal mixing efficiency. By harnessing the unique properties of these fluids at the nanoscale, such as their tunable viscosity and improved heat transfer capabilities, our research has demonstrated the potential for significant advancements in microscale mixing technology.

From this study, the following findings can be made:Within the suggested micromixer, Reynolds numbers have a greater impact on the hydrodynamic behavior of non-Newtonian nano-fluid.For any situation where there is a concentration of nano-fluid, the micromixer generates strong secondary flows to improve the mixing quality. When compared to the ideal scenario of a 5% nano-fluid concentration, the non-Newtonian fluid with a concentration of =0.5% shows a reduced mass transfer.The fluid flow mechanism of heat and mass transfer shows that the generation of the vortex generated within the micromixer has a powerful impact as the Reynolds number increases.Increased secondary flow rates in the micromixer have a greater effect on decreasing global entropy formation and increasing the thermal mixing degree.The concentration of the nano-fluid affects the generation of heat transfer entropy for all *Re*. As both *Re* and the flow behavior index (n) rise, so does the development of frictional entropy.In some applications where mixing efficiency is critical, the increased entropy generation seen in our studies suggests a higher level of disorder in the system, which can be advantageous.For the most ideal nano-fluid scenario in comparison to the other situations, the mean Bejan number decreases with increasing values of *Re*.

## Figures and Tables

**Figure 1 micromachines-15-01392-f001:**
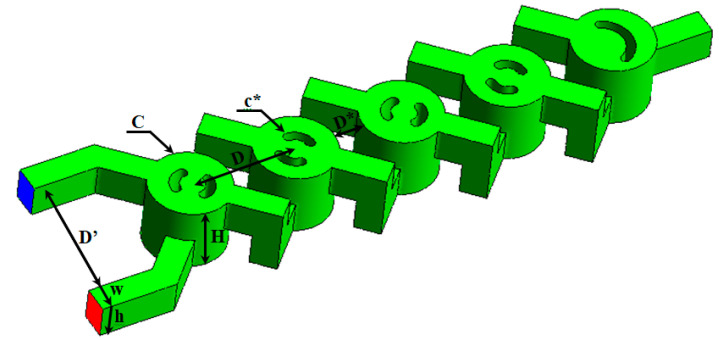
Two-layer crossing channels micromixer (TLCCM).

**Figure 2 micromachines-15-01392-f002:**
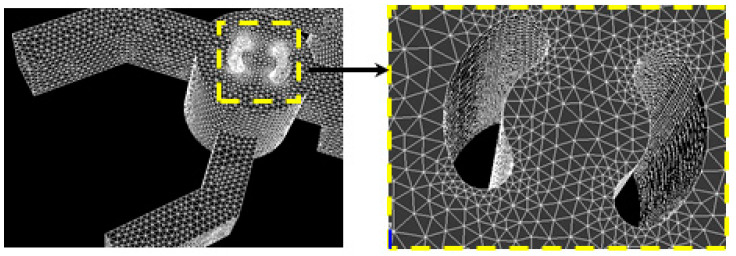
Capturing meshes.

**Figure 3 micromachines-15-01392-f003:**
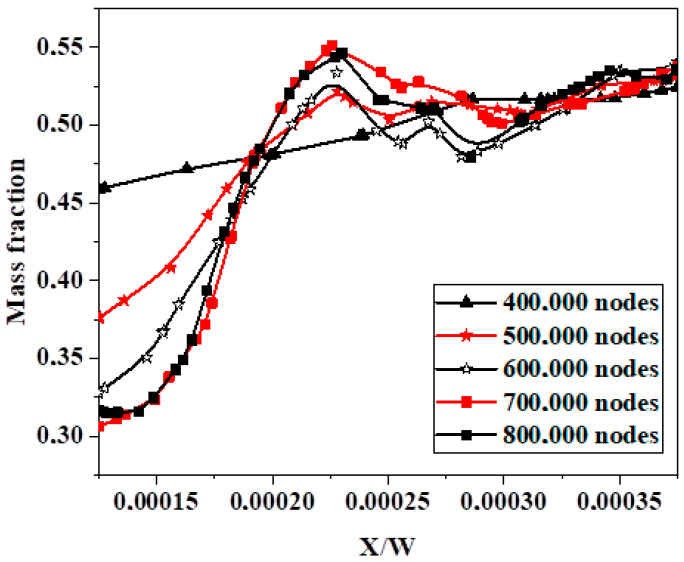
Mass fraction rate with dimensionless X-coordinates.

**Figure 4 micromachines-15-01392-f004:**
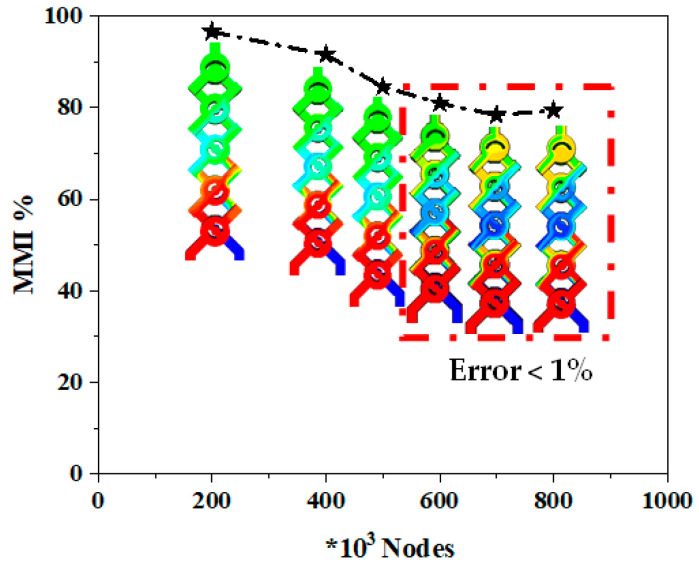
Mass mixing index for various nodes with mass fraction contours.

**Figure 5 micromachines-15-01392-f005:**
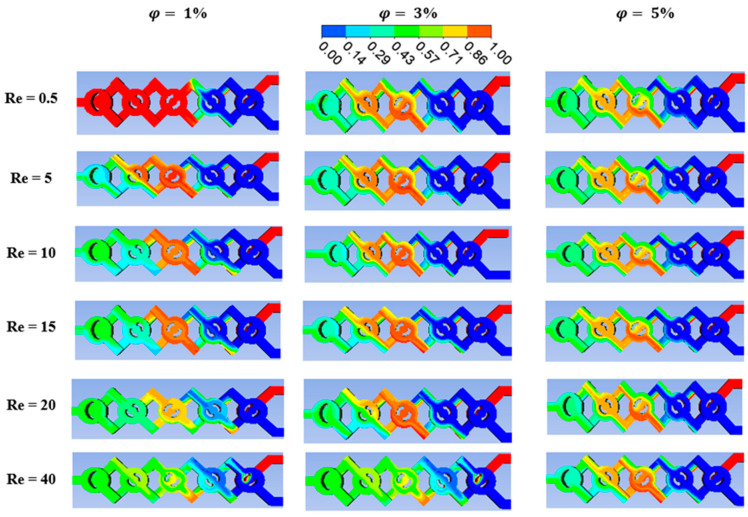
Mass fraction contours at the mid-cross section with different fluid concentrations at various Reynolds numbers (φ=1 to 5%).

**Figure 6 micromachines-15-01392-f006:**
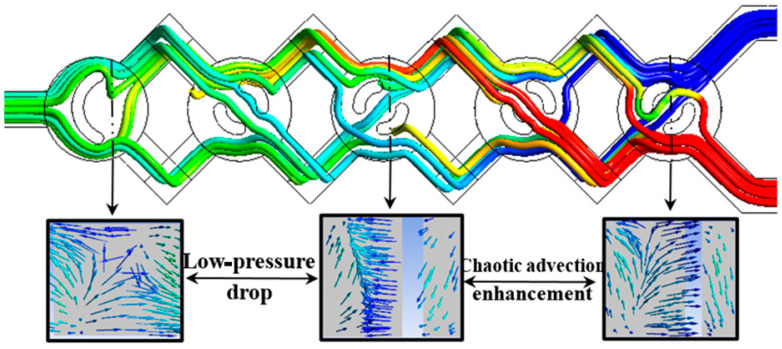
Vectors and streamlines of the mass fraction with φ=5% and Re=40.

**Figure 7 micromachines-15-01392-f007:**
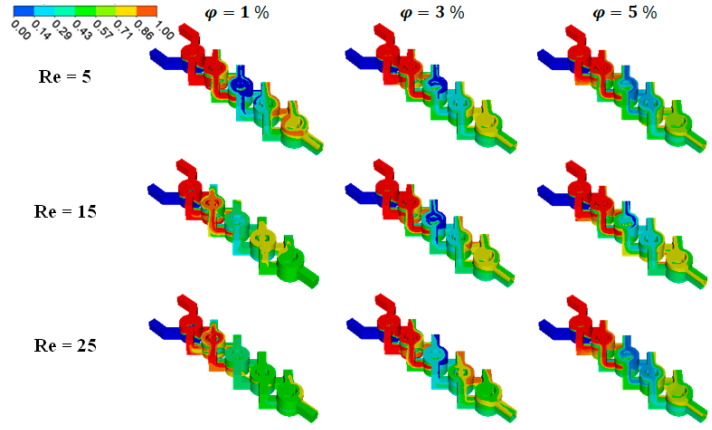
Mass fraction distributions at the outside micromixer with different fluid concentrations and Reynold numbers ranging from 0.5 to 25.

**Figure 8 micromachines-15-01392-f008:**
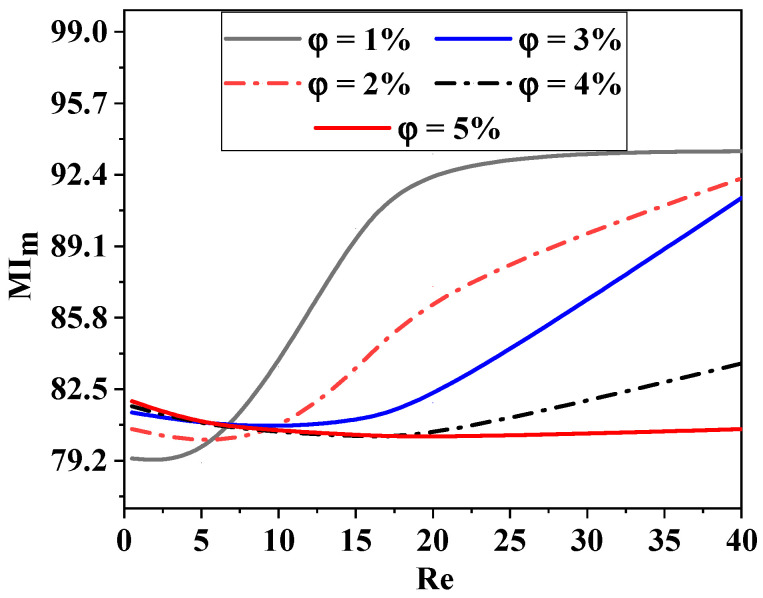
Improvement in mass mixing efficiency for varying Reynolds numbers and concentrations of nano-fluids (ϕ = 0.5 to 5%).

**Figure 9 micromachines-15-01392-f009:**
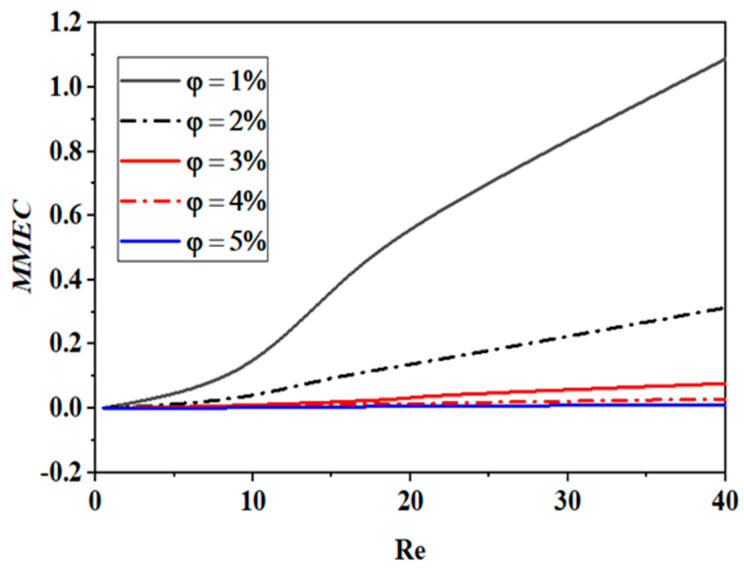
Growth in mass mixing energy cost for several Reynolds numbers with various nano-fluid concentrations (φ = 0.5 to 5%).

**Figure 10 micromachines-15-01392-f010:**
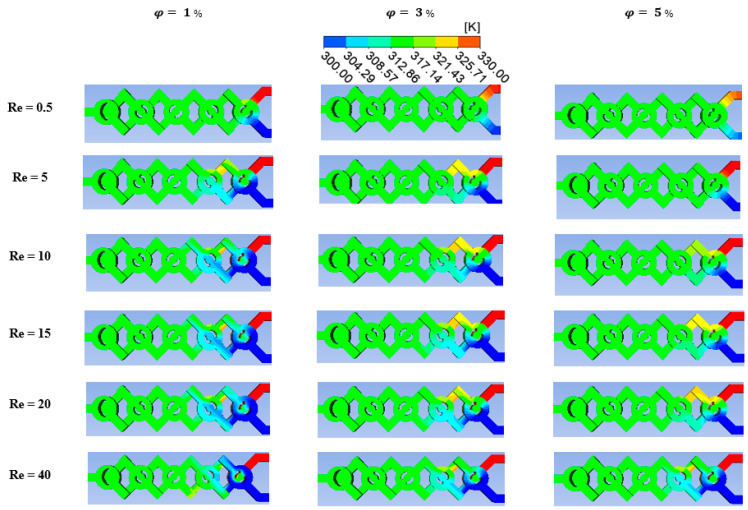
Temperature contours at the mid-cross section for various *Re* with several fluid concentrations, φ=1 to 5%.

**Figure 11 micromachines-15-01392-f011:**
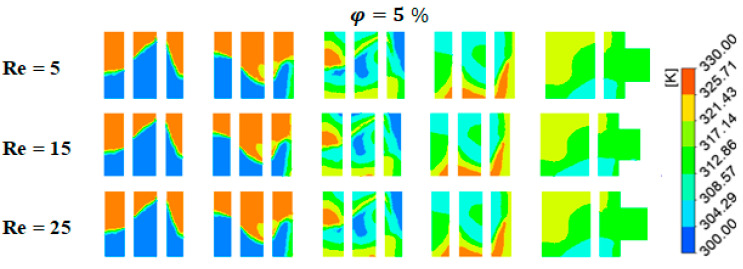
Temperature distributions at the middle cross section using various Reynolds numbers at fixed fluid concentrations and power-law indexes.

**Figure 12 micromachines-15-01392-f012:**
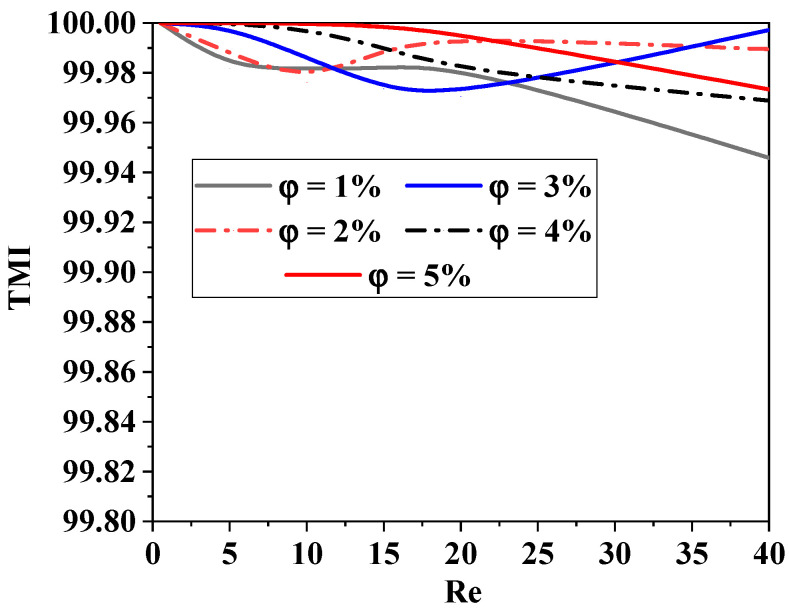
Improvement in thermal mixing performance for several Reynolds numbers with variant cases of nano-fluid concentrations (ϕ = 0.5 to 5%).

**Figure 13 micromachines-15-01392-f013:**
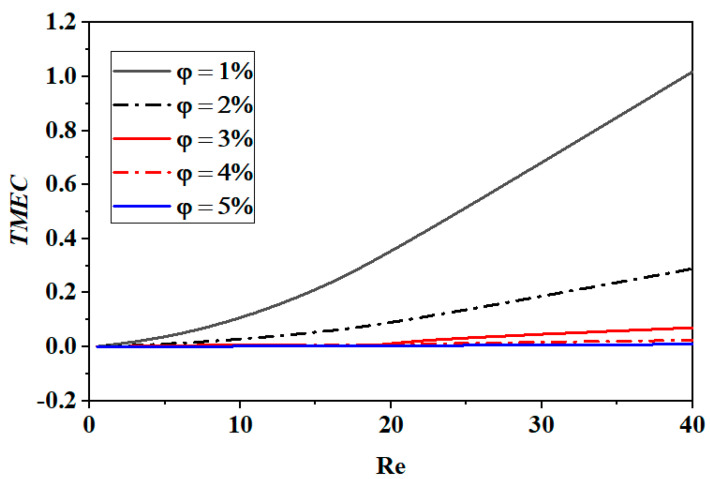
Growth in thermal mixing energy cost for numerous Reynolds numbers with different nano-fluid concentrations (φ = 0.5 to 5%).

**Figure 14 micromachines-15-01392-f014:**
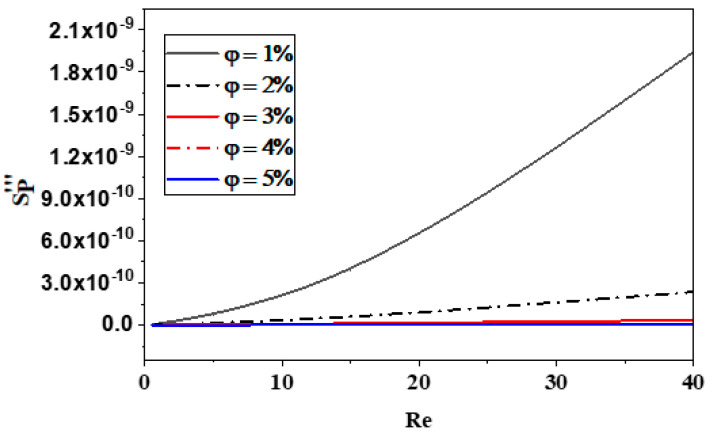
Effect of nano-fluid concentration on fluid friction entropy generation with various Reynolds numbers.

**Figure 15 micromachines-15-01392-f015:**
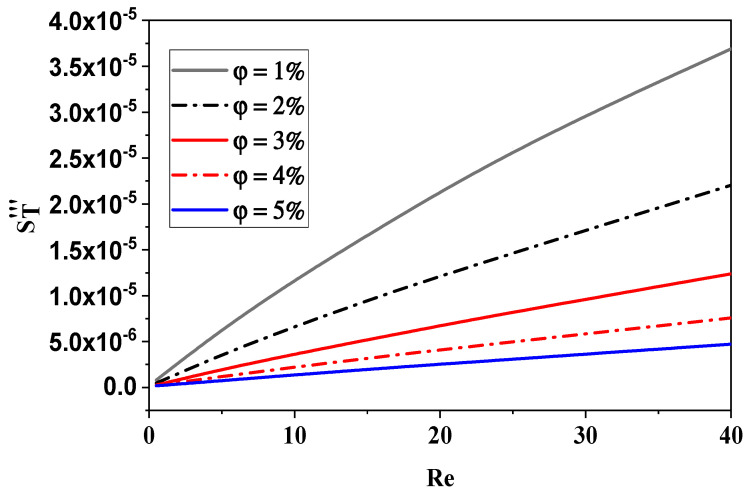
Effect of nano-fluid concentration on thermal entropy generation with various Reynolds numbers.

**Figure 16 micromachines-15-01392-f016:**
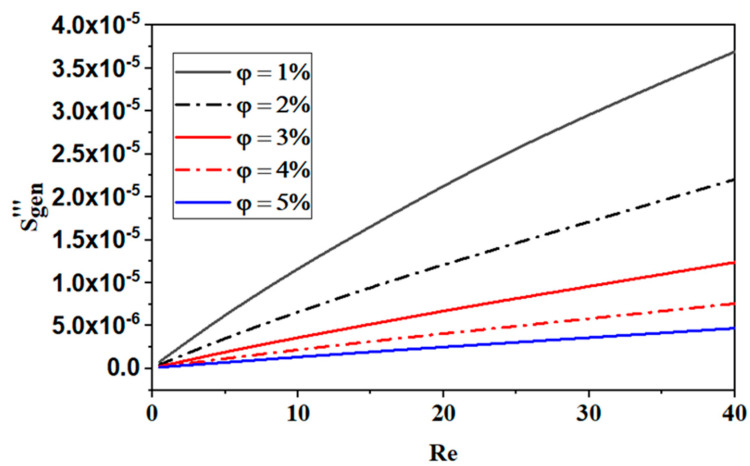
Effect of nano-fluid concentration on global entropy generation with various Reynolds numbers.

**Figure 17 micromachines-15-01392-f017:**
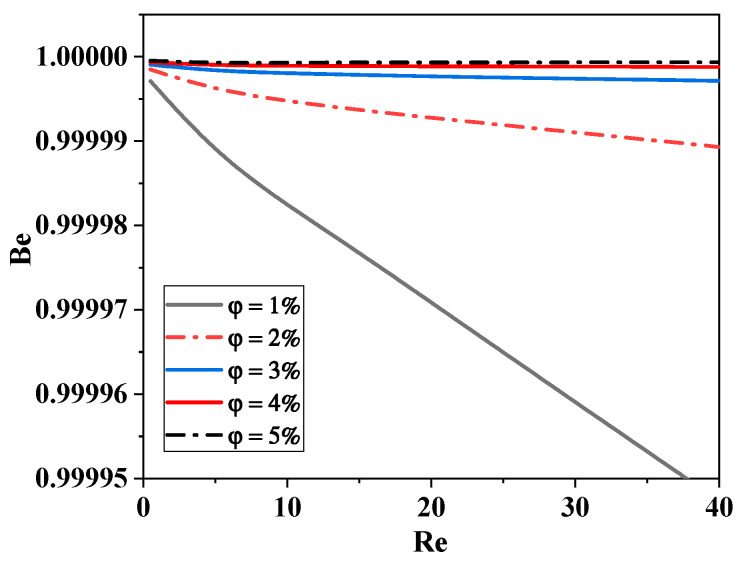
Effect of nano-fluid concentration on Bejan number with various Reynolds numbers.

**Table 1 micromachines-15-01392-t001:** Geometric parameters and their values.

Geometric Parameters	Values
W = h	0.2 mm
*D′*	0.8 mm
*D**	0.2 mm
*c**	0.1 mm
*d_hy_*	0.22 mm
*C*	0.6 mm
D	0.45 mm
H	0.4 mm

**Table 2 micromachines-15-01392-t002:** Non-Newtonian Al_2_O_3_ nano-fluid rheological parameters [30,31].

*Φ*%	m (Ns^n^m^−2^)	n
1.0	0.00230	0.83
2.0	0.00347	0.730
3.0	0.00535	0.625
4.0	0.00750	0.540
5.0	0.01020	0.460

**Table 3 micromachines-15-01392-t003:** Mesh sensitivity for the pressure drop and standard deviation of fluids.

Mesh Nodes	Pressure Drop (Pa)	Standard Deviation of Fluid
≈200.000	204.5	0.063
≈300.000	206	0.068
≈400.000	206	0.072
≈500.000	209.5	0.076
≈600.000	211.3	0.078
≈700.000	211.4	0.078
Error %	0.004	0.001

**Table 4 micromachines-15-01392-t004:** Comparison of mixing efficiency with Lee et al. [5].

*Re* = 8	Mixing Efficiency %	
Quantity of Obstacles	5	8
Experimental of Lee et al. [5]	60.80	82.07
Present simulation.	60.72	81.38
Error	0.13%	0.84%

**Table 5 micromachines-15-01392-t005:** Reynolds number vs. heat transfer coefficient for a non-Newtonian scenario (n = 2).

*Re*	Present Work [w/m^2^.k]	Li et al. [64] [w/m^2^.k]	Error
15	19,958	16,643	0.199
20	24,092	21,284	0.131
90	48,311	49,130	0.016
110	55,761	56,600	0.014
220	82,456	82,963	0.006

**Table 6 micromachines-15-01392-t006:** Assessment of vortex strength for different Reynolds numbers using various nano-fluid cases.

*Re*	φ = 1%	φ = 2%	φ = 3%	φ = 4%	φ = 5%
5	12.457	0.0607	3.614	2.143	1.270
10	98.058	53.361	28.261	16.633	9.985
15	184.272	99.613	52.511	30.862	18.525
20	268.217	143.478	98.120	44.378	26.622
25	352.203	187.359	120.490	57.549	34.494

**Table 7 micromachines-15-01392-t007:** Comparison of mixing energy cost of micromixers.

*Re*	MMEC [65]	MMEC [66]	Present MMEC
1	0.03	0.0013	0.0013
5	0.834	0.336	0.194
15	10.26	2.456	1.548
25	51.46	11.529	7.25

## Data Availability

The original contributions presented in the study are included in the article, further inquiries can be directed to the corresponding author.

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
