# Peer review of "Analysis of Entropy Generation for Mass and Thermal Mixing Behaviors in Non-Newtonian Nano-Fluids of a Crossing Micromixer"

_micromachines, 2024, doi:10.3390/mi15111392_

Round 1

Reviewer 1 Report

Comments and Suggestions for Authors

The manuscript presents a study on the thermal and mass mixing behaviors of non-Newtonian nanofluids in a Two-layer Crossing Channels Micromixer (TLCCM), emphasizing entropy generation and energy efficiency. This research provides valuable insights into the optimization of micromixers using nanofluids, particularly at low Reynolds numbers.

My specific comments and suggestions are as follows:

1. The introduction provides a broad overview of micromixers and their applications. However, I recommend emphasizing the distinctiveness of the proposed TLCCM design and how it advances beyond the existing literature. For instance, clarifying the novelty of combining specific non-Newtonian fluids with the TLCCM design will enhance the research’s uniqueness and importance.

2. While the study uses CFD simulation effectively, additional details on the CFD setup, including mesh parameters, boundary conditions, and grid sensitivity analysis, would be beneficial. This information would improve reproducibility and give readers greater confidence in the model's validity. These details are essential for assessing the accuracy and robustness of the computational model.

3. further explanation of how various Reynolds numbers and nanoparticle concentrations affect mixing efficiency and energy cost would provide clearer insights. Adding a comparative table or graph would aid readers in understanding these variations.

4. Entropy generation is a central theme in this paper. To enhance the practical application, I suggest discussing how entropy minimization could impact energy efficiency in specific applications, such as chemical processing or biomedical engineering.

5. The conclusion summarizes key findings, but it would benefit from suggestions for future research directions, such as potential experimental validations or applications in various fields. This addition would make the manuscript more comprehensive and encourage further exploration of the findings.

Reviewer 2 Report

Comments and Suggestions for Authors

Review of the paper

 Analysis of Entropy Generation for Mass and Thermal Mixing  Behaviours in Non-Newtonian Nano-fluids of a crossing Micromixer

A Two-layer Crossing Channels Micromixer (TLCCM) has been studied in the paper.

For a range of Reynolds numbers from 0.1 to 25, the impact of fluid characteristics and various concentrations of Al2O3 nanoparticles on the 25 thermal mixing capabilities and pressure drop were investigated.

1) The authors use the term “the strong chaotic flow”, firstly in abstract.

For the quantitative analysis there should be some parameter defined which reflects the amount of chaotic flow, in order to assess this property exactly.

2) The authors have studied quite narrow range of Reynolds numbers – just from 0.1 to 25.

Obviously, the increase of Reynolds number up to 1000—2000 at least will result in much higher mixing behaviour of the device.

Is the any data related to larger Reynolds numbers?

3) The dimensions presented in Fig. 1 and in Table 1 are not sufficient to produce similar device by other researcher groups. Please make this outline more precise. E.g. no one of height values is presented here.

4) Table 2. There is “N” term, but there is no “n” – exponent of Eq. 4.

5) Reference 54. The first author’s name is “A. B. Metzner”. Please correct this.

6) Reference 48. The full description of the paper is missing, there is just doi.

7) Fig. 2 was already presented by authors in their recent paper [24] Naas.T.T, Shakhawat.H ,Abid.H.K, Telha.M andKwang-Yong.K, Evaluation of Hydrodynamic and Thermal Behaviour 501 of Non-Newtonian-Nano-fluid Mixing in a Chaotic Micromixer, Micromachines 2022, 13, 502 933.https://doi.org/10.3390/mi13060933.

It is not allowed to publish the same plot, especially without reference to the original paper.

8) Continuation of comment 7. The novelty of this manuscript should be explicitly shown in the Introduction section, related to the paper [24].

 9) Fig. 4. Again, some quantitative parameter is necessary to evaluate the level of chaotic advection. Is there any variable for such assessment?

10) The same comment is applicable for Fig. 5. There are a lot of similar pictures, but a quantification of the mass transfer is missing here. The mixing index values would be appropriate here.

11) Fig. 6, 7, 10, 11 and many other are drawn without any point! Please add them.

12) Fig. 9 looks very strange: 24 similar green rectangles!

13) No one of plots have any estimation of the experimental error. Taking into account that the results are based on the numerical simulations only, their correctness should be proven experimentally.

Numerical without experimental proof, especially for fluids with nanoparticles, have very low impact.

That is why I cannot recommend this paper for publication.

Round 2

Reviewer 2 Report

Comments and Suggestions for Authors

The authors have fixed all the issues raised by the reviewer.

The paper could be recommended now for publication